# Isotopic Composition of Glacier Ice and Meltwater in the Arid Parts of the Altai Mountains (Central Asia)

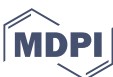

**Dmitriy Bantcev** [1,*], **Dmitriy Ganyushkin** [1], **Anton Terekhov** [2], **Alexey Ekaykin** [1,3], **Igor Tokarev** [4] and **Kirill Chistyakov** [1]

1   Institute of Earth Science, Saint-Petersburg State University, Universitetskaya nab. 7/9, 199034 Saint-Petersburg, Russia; d.ganyushkin@spbu.ru (D.G.); ekaykin@aari.ru (A.E.); k.chistyakov@spbu.ru (K.C.)
2   State Hydrological Institute, Vasilyevsky Island, 2nd Line, 23, 199004 Saint-Petersburg, Russia; antonvterekhov@gmail.com
3   Arctic and Antarctic Research Institute, Beringa 38, 199397 Saint-Petersburg, Russia
4   Research Park, Saint-Petersburg State University, Universitetskaya nab. 7/9, 199034 Saint-Petersburg, Russia; i.tokarev@spbu.ru
*   Correspondence: d.bantcev@spbu.ru

**Abstract:** The objective of this study is to reveal the isotopic composition of ice and meltwater in glaciated regions of South-Eastern Altai. The paper depicts differences between the isotopic composition of glacier ice from several types of glaciers and from various locations. Detected differences between the isotopic composition of glacier ice in diversified parts of the study region are related to local climate patterns. Isotopic composition of meltwater and isotopic separation for glacier rivers runoff showed that in the Tavan-Bogd massif, seasonal snow participates more in the formation of glacier runoff due to better conditions for snow accumulation on the surface of glaciers. In other research areas pure glacier meltwater prevails in runoff.

**Keywords:** glaciers; runoff; isotopic composition; isotopic separation; Altai

## 1. Introduction

The use of environmental isotope tracers (such as isotopic composition of natural waters) for the solution to hydrological and glaciological problems is one of the leading directions in the development of Earth sciences. Isotope studies in high mountain regions may be used for further research of glaciers features and mountain river runoff.

The South-Eastern Altai is characterized by arid climate and is a problem area in terms of water resources. Due to the small amount of precipitation runoff from glaciers in the South-East, Altai plays a special role in the economic activity of the local population. The study of high mountain glaciers and rivers for water balance research is a cornerstone for water management estimates and forecasting, especially within the framework of global climate change.

To achieve this aim, the following tasks were solved: evaluation and comparison of the isotopic characteristics of the glacier's ice in different parts of South-Eastern Altai and the evaluation of the isotopic composition and its time variability for the water of glacial rivers.

In Russia, stable isotopes research of glacier river runoff and glacier ice has been carried out for several years in the Central Caucasus, using examples of runoff from the Dzhankuat glacier (mainly by the Moscow State University) [1–3].

Isotopic research in the Altai mountains in general demands deep ice drilling. Earlier the ice core was investigated on Belukha Mountain [4,5]. The ice core was also sampled at the Tsambagarav massif (Mongolia) [6]. Apart from this ice-water, isotopic features and relations in this study area are not well presented.

The isotopic composition of water has been studied thoroughly in areas closest to the Altai Mountain areas, which are located in China. Earlier studies have determined

the prevalence of glacial runoff in the total runoff from July to August and the prevalence of ground feeding in winter for the Urumqi River [7]. Similar studies of the isotopic composition of glacial-originating rivers were also performed in the north of the Tibetan mountains in China. The authors determined the isotopic characteristics of the components, which form the runoff, including snow, firn and glacial ice, and evaluated the role of glaciation systems in the feeding of the studied rivers [8–11]. Another isotopic separation study was carried out in a glacierized region of Southwest China [12]. In this part of China, the Lancang River has also been the object of complex stable isotopes research [13].

As stated above, Russian and Mongolian parts of the South-Eastern Altai are not well-provided by glacier ice and glacier runoff isotopic research. Additionally, this kind of research can be very useful especially within the framework of global climate change and further deglaciation.

## 2. Materials and Methods

Sample collection was carried out in the middle of the ablation season, from July to August. Over the course of nine years, more than 800 samples were drawn. Glacial ice was collected from the surface of glaciers into the plastic bags, then melted at an ambient temperature and poured into test tubes. Samples from watercourses were drawn directly into test tubes. Precipitation samples were collected immediately after precipitation to avoid evaporation

The relative isotopic content is indicated by $\delta$. SMOW (standard mean ocean water) is taken as the zero standard.

$$\delta^{18}O = [(^{18}O/^{16}O)sample - (^{18}O/^{16}O)SMOW]/(^{18}O/^{16}O)SMOW \times 10^3$$

$$\delta^2H = [(D/H)sample - (D/H)SMOW]/(D/H)SMOW \times 10^3$$

The analysis of isotopic characteristics for samples collected in 2012–2017 was carried out at the Laboratory for Climate and Environmental Change at the Arctic and Antarctic Research Institute with Cavity Ring-Down Spectroscopy (CRDS) using a a Picarro L2120-i gas analyser. Distilled tap water from St. Petersburg was used as a laboratory standard; it had the following characteristics: $-9.79‰$ in $\delta^{18}O$ and $-75.47‰$ in $\delta^2H$ relative to the IAEA "V-SMOW2" standard. The measurement precision was 0.05‰ for $\delta^{18}O$ and 0.5‰ for $\delta^2H$, which is quite enough for this kind of research. Isotopic composition was rounded off to the nearest 0.1. In 2018, analysis was fulfilled at the Resource Center "X-ray Diffraction Research Methods" of the St. Petersburg State University Science Park using the same spectrometer. USGS45, USGS46 and GISP standards were used with the same measurement accuracy.

The relationship between the relative content of $^{18}O$ and D in precipitation was determined by the fractionation coefficients of these isotopes. The formula of linear relationship between $^{18}O$ and D was obtained by H. Craig, and it has the following form:

$$\delta^2H = 8\delta^{18}O + 10$$

This linear relation is designated as global meteoric water line (GMWL) [14].

In 2005, this equation was refined using long-term means from GNIP stations and the resulting GMWL was [15]:

$$\delta^2H = (8.14 \pm 0.02) \times \delta^{18}O + (10.9 \pm 0.2), R = 0.98$$

Studies of runoff in areas with developed glaciation are based on the fact that using stable isotopes as tracers, it is possible to divide the hydrograph of glacial rivers into components. The use of stable isotopes in glaciology and hydrology is based on natural differences in the isotopic composition of the components of the glacial rivers' runoff. Isotopic methods allow to carry out only two-component separation, i.e., to calculate the shares of the two most significant components in the total runoff can be calculated.

This separation is carried out using the isotope balance formula, which generally has the form:

$$R^{18}O_1f_1 + R^{18}O_2f_2 = R^{18}O$$

with $R^{18}O_1$—isotopic content of the first component, $f_1$—share of the first component, $R^{18}O_2$—isotopic content of the second component, $f_2$—share of the second component, $R^{18}O$—resulting isotopic composition [1].

## 3. Results

### 3.1. Isotopic Composition of Glacier Ice

Glaciers are one of the main water sources in arid South-Eastern Altai. To estimate their role in alimentation of mountain rivers, we required data on the isotopic composition of glacier ice. Sampling of glacier ice was carried out in three massifs: Tavan-Bogd, Mongun-Taiga in Russia and Tsambagarav in Mongolia (Figure 1).

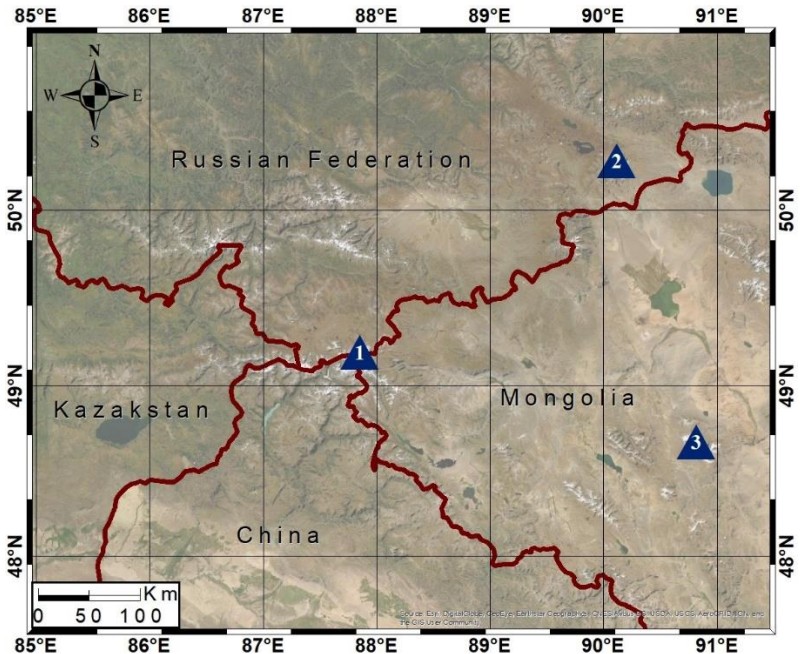

**Figure 1.** Sampling areas: 1—Tavan-Bogd; 2—Mongun-Taiga; 3—Tsambagarav.

The Tavan-Bogd massif is a large centre of modern glaciation. The northern slope of the massif is located in Russia. The south-eastern part belongs to Mongolia and the south-western part belongs to China. The northern slope of the Tavan-Bogd massif is characterized by smaller areas of glaciation. In 2015 there were 16 glaciers on its territory with a total area of 23.46 km². The average height of the firn border is 3335 m. One of the features of this massif is an increase in precipitation from east to west. The annual amount of precipitation in the highlands of the massif extends from 364 mm in the eastern part of the massif to 880 mm in the western part. The average height of the firn border decreases from 3415 m in the east to 3150 m in the west of the massif [16].

The Mongun-Taiga massif is located in the far east of Altai. The distribution of heat and moisture varies due to the significant height of the massif and the complexity of its relief. Most precipitation falls on the windward slopes and in the high-altitude zone. In 2010, there were 32 glaciers in the massif with a total area of 20.3 km² (The Mongun-Taiga Mountain Massif, 2012). The average height of the firn border is 3390 m. The amount of precipitation on it was estimated at 269 mm [17].

Tsambagarav massif is located in the north-west of Mongolia. In 2015, there were 67 glaciers on the ridge with a total area of 68.4 km²; the weighted average height of the firn border is 3748 m. There are large valley and cirque-valley glaciers on the north-eastern

slope, the tongues of the largest of them descend to a height of 3000 m. The firn border on the glaciers of the north-eastern slope of Tsambagarav is located at an altitude 3500–3750 m.

Most of the samples were drawn in the ablation zones of valley glaciers. In the Mongun-Taiga massif, samples of ice from corrie glaciers and from the glacier dome on the top (3976 m.e.s.l.) were collected. Isotopic content of ice from different types of glaciers has noticeable differences. Box plots for $\delta^{18}O$ in glacier ice from different types of glaciers are displayed at Figure 2. Mean $\delta^{18}O$ values are also shown in Table 1.

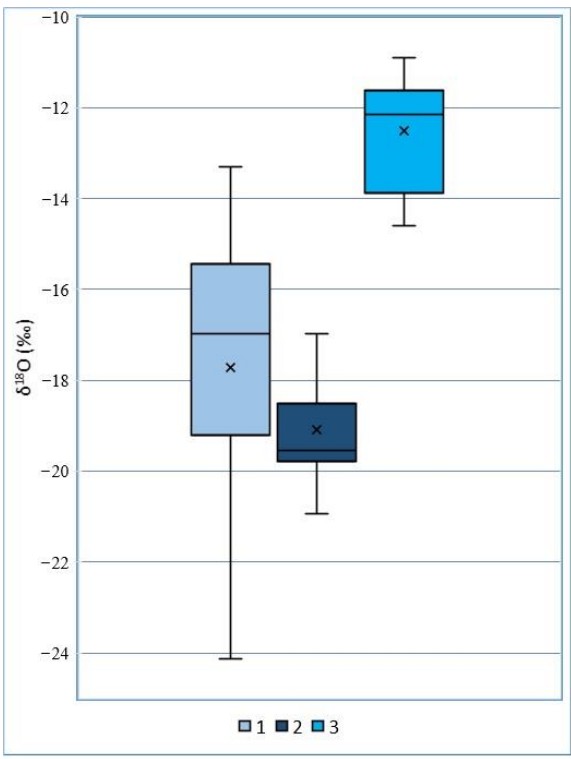

**Figure 2.** Box plots for $\delta^{18}O$ in glacier ice of valley (1), corrie (2) glaciers and glacier dome (3) at Mongun-Taiga massif.

**Table 1.** Mean $\delta^{18}O$ values for different types of glaciers at Mongun-Taiga massif.

| Type | Number of Samples | Mean $\delta^{18}O \pm$ 2SEM |
|---|---|---|
| Valley | 49 | $-17.7 \pm 0.9$ |
| Corrie | 20 | $-19.1 \pm 0.9$ |
| Dome | 26 | $-12.5 \pm 0.5$ |

A maximum range of $\delta^{18}O$ is observed in the ice samples from the ablation zones (ice tongue) of the valley glaciers. For corrie glaciers, mean $\delta^{18}O$ values are lower, and the range is minimal. The samples of the glacier dome differ from other groups of samples significantly: the mean value is higher and the range of $\delta^{18}O$ is not as large as the ranges from valley glacier samples. The reason of these differences will be discussed in part 4.

In Figure 3, box plots for $\delta^{18}O$ of valley glaciers ice in three research areas are presented. Mean values are also presented in Table 2. The isotopic composition of glacier ice in the Tsambagarav massif was analysed more in a previous manuscript [18]. In Figure 3 and in Table 2, an extra 27 samples from Tsambagarav, which were collected in 2019, were added to the statistics.

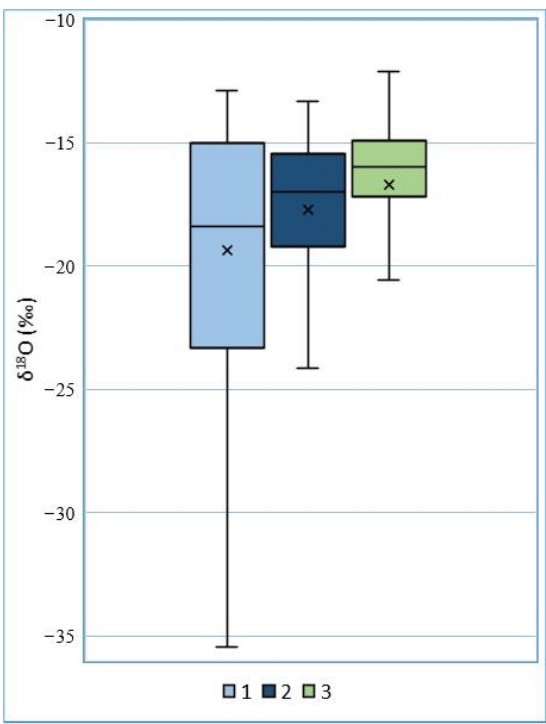

**Figure 3.** Box plots for $\delta^{18}O$ in valley glaciers ice in the Tavan-Bogd (1), Mongun-Taiga (2) and Tsambagarav (3) massifs.

**Table 2.** Mean $\delta^{18}O$ values for valley glaciers in 3 research areas.

| Massif | Number of Samples | Mean $\delta^{18}O \pm$ 2SEM |
|---|---|---|
| Tavan-Bogd | 29 | −19.4 ± 2.1 |
| Mongun-Taiga | 49 | −17.7 ± 0.9 |
| Tsambagarav | 78 | −16.7 ± 0.8 |

The range of isotopic composition decreases and the mean values of $\delta^{18}O$ increase from Tavan-Bogd to Tsambagarav.

These isotopic characteristics of ice can be used in meltwater and river runoff research. In the next chapter, the isotopic characteristics of glacier-originated water will be presented.

### 3.2. Isotopic Composition of Glacier Meltwater

Most of the meltwater samples were taken in the Tavan-Bogd massif. Sampling was carried out in 2015 [19], 2018 and 2021 in the middle of ablation season. Meltwater samples were taken from the edges of most glaciers on the northern slope (Russian part of this massif). Snow and firn samples were also collected from the glacier surface for further isotopic separation.

Figure 4 illustrates the scheme of the glaciers on the northern slope of Tavan-Bogd.

The isotopic composition of the meltwater from the main glaciers is displayed in Table 3. Isotopic composition of the meltwater from the two largest glaciers (No. 11 and No. 14 at Figure 4) did not change significantly. The meltwater from glacier No. 3 was also rather constant. Isotopic composition of meltwater from other glaciers varied more noticeably and it was in general higher in 2015 and 2021, but lower in 2018. During the sampling in 2018 most of the glaciers in the central part of the massif were still covered with snow with low isotopic composition. This fact was reflected in the lower $\delta^{18}O$ values.

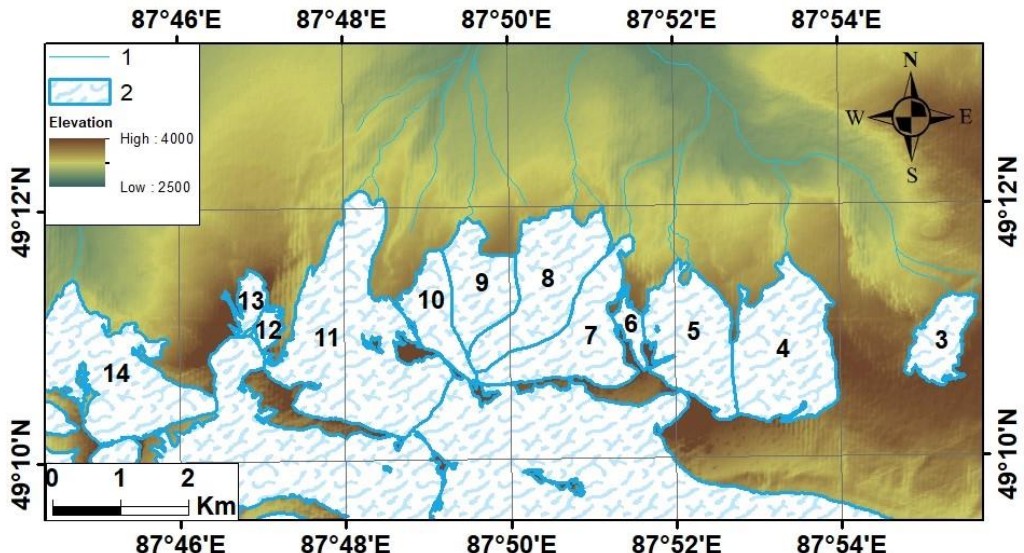

**Figure 4.** Glaciers of the northern slope of the Tavan-Bogd massif. 1—Rivers; 2—Glaciers.

**Table 3.** Meltwater $\delta^{18}$O for glaciers at the northern slope of the Tavan-Bogd massif.

| No. of Glacier | $\delta^{18}$O 2015 | $\delta^{18}$O 2018 | $\delta^{18}$O 2021 |
|---|---|---|---|
| 3 | −17.7 | - | −17.6 |
| 4 | −13.9 | −16.5 | - |
| 5 | −14.4 | −17.3 | −15.2 |
| 7 | −16.2 | −17.5 | −15.9 |
| 8 | −14.9 | −17.3 | −15.4 |
| 9 | −13.8 | −16.2 | −15.7 |
| 10 | −14.5 | - | −14.7 |
| 11 | −15.9 | −15.9 | −15.7 |
| 14 | −16.5 | −16.7 | −15.8 |

These isotopic differences are related with morphometric types of glaciers and features of accumulation, and these relations will be discussed in part 4 of this study.

Meltwater samples have also been collected in other research areas. The isotopic composition of the runoff in the Tsambagarav massif was analysed in previous work [18]. In the Mongun-Taiga massif meltwater $\delta^{18}$O varies from −14.8‰ to −17.9‰, but most of meltwater samples have $\delta^{18}$O ‰ close to −16‰.

Diagrams of $\delta^{18}$O-$\delta^2$H relations for ice and meltwater samples (Table S1) for all research areas are presented below (Figure 5, Table 4). The Tavan-Bogd massif meltwater samples, which were collected in 2018, are not included due to abnormal conditions that year (ice tongues of most of the glaciers were covered by snow even in the middle of summer).

**Table 4.** $\delta^{18}$O-$\delta^2$H relations equations for ice and meltwater.

| Massif | Equation for Ice Samples | Equation for Meltwater Samples |
|---|---|---|
| Tavan-Bogd | $\delta^2$H = 8.07 $\delta^{18}$O + 18.54, R$^2$ = 0.99 | $\delta^2$H = 7.60 $\delta^{18}$O + 8.69, R$^2$ = 0.95 |
| Mongun-Taiga | $\delta^2$H = 7.75 $\delta^{18}$O + 12.19, R$^2$ = 0.99 | $\delta^2$H = 7.67 $\delta^{18}$O + 8.83, R$^2$ = 0.98 |
| Tsambagarav | $\delta^2$H = 7.74 $\delta^{18}$O + 11.16, R$^2$ = 0.99 | $\delta^2$H = 7.46 $\delta^{18}$O + 6.46, R$^2$ = 0.99 |

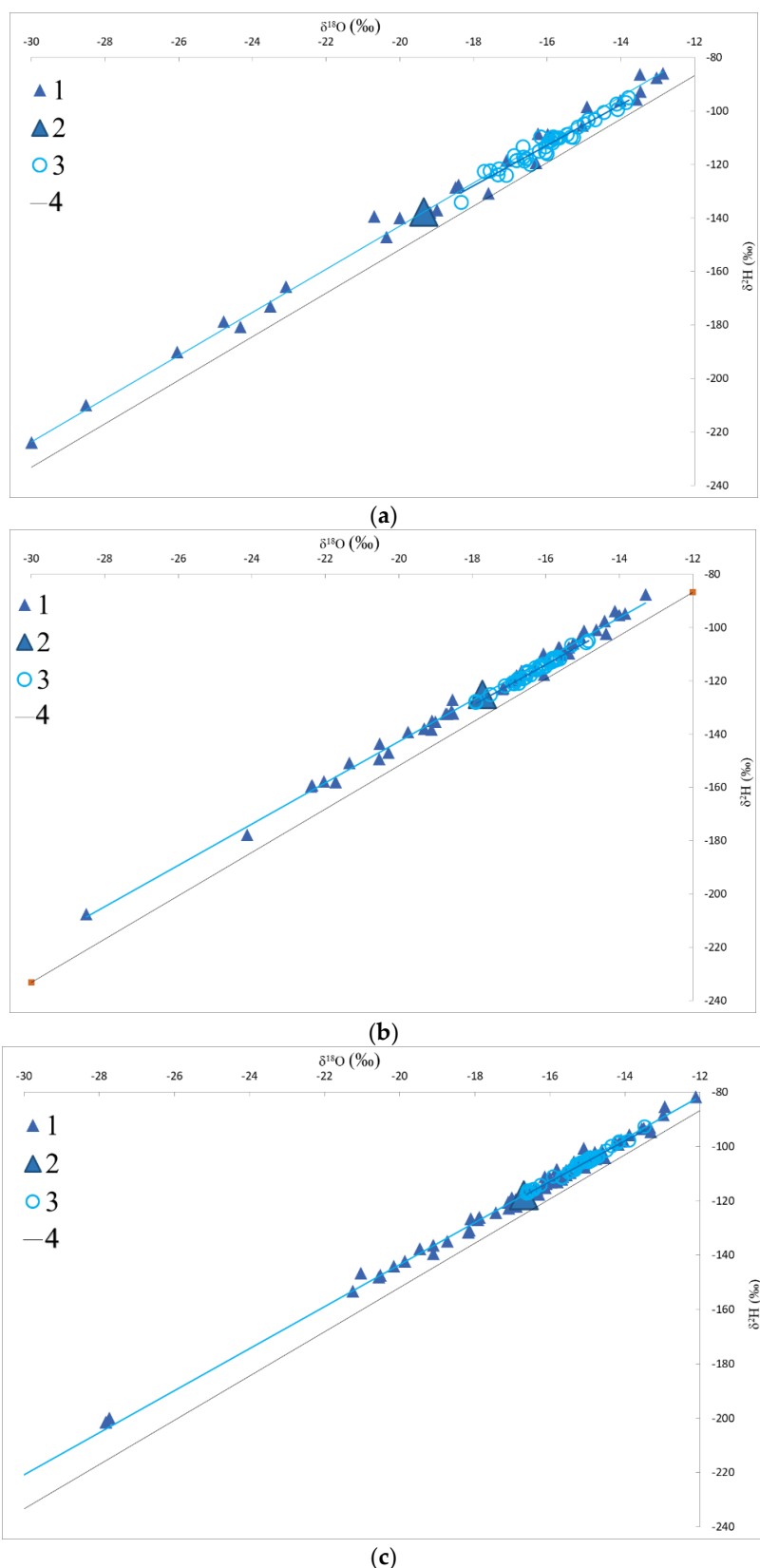

**Figure 5.** $\delta^{18}O$-$\delta^{2}H$ relations for ice and meltwater in the Tavan-Bogd (**a**), Mongun-Taiga (**b**) and Tsambagarav (**c**) massifs. 1—ice from valley glaciers; 2—mean value of the ice samples; 3—meltwater; 4—GMWL ($\delta^{2}H = 8.14\ \delta^{18}O + 10.9$).

As Figure 5 illustrates, meltwater from the Tavan-Bogd (Figure 5a) is rather far from the mean ice value in the comparison to other two massifs.

The slope of the regression line between $\delta^{18}O$ and $\delta^2H$ for Tavan-Bogd samples is close to GMWL ($\delta^2H = 8.14\delta^{18}O + 10.9$). The slope of the regression for Mongun-Taiga and Tsambagarav samples is close to meteoric water line in Arid Central Asia ($\delta^2H = 7.67 \delta^{18}O + 3.32$) [20] due to the higher aridity in these massifs.

## 4. Results

### 4.1. Differences in the Isotopic Composition from Different Types of Glaciers in the Mongun-Taiga Massif

On the top of the glacier dome, summer snow with high $\delta^{18}O$ melts and refreezes on the surface of glacier. This is why $\delta^{18}O$ of samples from the dome, which were taken in the middle of summer, were maximal and the range of $\delta^{18}O$ values was minimal.

Valley glaciers have large accumulation zones, which include corries and cirques where winter snow concentrates. That is why $\delta^{18}O$ in valley glacier ice has the largest range. It is possible that the mean $\delta^{18}O$ value of valley glacier ice reflects the mean isotopic characteristics of the accumulated precipitation. Valley glaciers play the main role in the feeding of the glacier rivers, so this value was used in further isotopic separation.

The mean isotopic content of the corrie glacier ice is lower, but close to the valley glaciers. The range is minimal. Minimal $\delta^{18}O$ can be explained by better conditions for the accumulation of precipitation during cold seasons. Minimal range can be explained by water refreezing predominance according to small size of this kind of glacier. However, small number of samples (20) does not allow to make any strong conclusions.

### 4.2. Differences in Isotopic Composition of Valley Glaciers Ice from Different Research Areas

The maximum range of $\delta^{18}O$ values and the minimum average $\delta^{18}O$ is observed for ice samples on the territory of the Tavan-Bogd massif. Increase in average values and decrease in the $\delta^{18}O$ range in glacial ice samples in the rest of the massifs is associated with an increase in aridity and, as a result, with a decrease in annual precipitation and an increase of summer precipitation share in accumulation (Table 5).

**Table 5.** Precipitation amount and share of the summer precipitation in research areas.

| Massif | Precipitation Amount on the Firn Border, mm | Share of the Summer Precipitation, % |
|---|---|---|
| Tavan-Bogd | 364–660 | 80 |
| Mongun-Taiga | 269 | 81 |
| Tsambagarav | 270 | 89 |

Since the Tavan-Bogd massif is located to the west from the others, more precipitation falls on its territory due to the predominance of westerlies in the Altai mountains. Despite the predominance of summer precipitation typical for the whole territory of arid highlands, winter precipitation with low isotopic composition plays a greater part in the formation of ice here than on the other massifs. This is reflected by larger range of values and a minimum mean value of $\delta^{18}O$. Summer precipitation falls mainly in liquid form and weakly participates in the formation of ice. Winter precipitation occurs more often than in other studied massifs, and, therefore, is more involved in the accumulation, which is reflected in its isotopic composition.

The ice of the Tsambagarav massif glaciers, which is located to the southeast of the other sites and exists in the most arid conditions, has higher isotopic composition with a minimum range of $\delta^{18}O$. This indicates higher a contribution of precipitation during the warm season, since winter precipitation is practically absent.

The mean and median $\delta^{18}O$ values of glacial ice of the valley glaciers in the Mongun-Taiga massif occupy an intermediate position between the glaciers of the Tsambagarav and

Tavan-Bogd massifs. For both the Tsambagarav massif and the Mongun-Taiga massif, the $\delta^{18}$O values of most glacial ice samples are within the range of $-15$ to $-20$‰.

Comparison with OIPC data (model data about isotopic composition of precipitation, which is based on GNIP stations interpolation) shows that the ice of the glaciers of these massifs is formed primarily by the precipitation of the transitional seasons (autumn-spring), but for the Tsambagarav massif, as was noted above, precipitation of the warm season plays a slightly larger role. For the Tavan-Bogd massif, this statement is less relevant [21,22]. Thus, the isotopic composition of the glacial ice of the arid highlands glaciers reflects well the conditions for the accumulation of precipitation, from which these glaciers form.

*4.3. Meltwater Isotopic Composition*

The glaciation of the northern slope of Tavan-Bogd is represented primarily by small corrie glaciers. The isotopic composition of the meltwater shows that in the middle of the ablation season, snow from the surface of glaciers plays an important role in the glacial rivers feeding in the northern part of the massif, in contrast to the rivers of the other massifs. It is explained by the predominance of morphological types of glaciers, which are favourable for snow accumulation. The ratio of the snow and ice components in the glacial runoff varies depending on the size and morphological type of the glacier. For most small corrie glaciers with favourable snow accumulation conditions, the proportion of melt snow water in the glacial runoff is greater than in the valley. This is observed both in the values of the isotopic composition of meltwater and in its variability over the years: Table 3 shows that meltwater $\delta^{18}$O was almost constant in 2015, 2018 and 2021 for valley glaciers and changed from year to year for the most of other glaciers. The sampling in 2018 was carried out for 10 days near glacier No. 11 and the glacier No. 11 and meltwater $\delta^{18}$O was also constant. Lower values of $\delta^{18}$O also display predominance of glacier meltwater in total runoff.

However, for the non-valley glaciers share of snow meltwater in total runoff is more significant. Thus, the ability of glaciers to accumulate large masses of spring snow on their surface is their important hydrological role on the northern slope of Tavan-Bogd.

During detailed glaciological observations that have been conducted on the territory of the massif since 1999, a tendency toward a rapid retreat of the valley glaciers was observed in this area. So, it can be assumed that the role of seasonal snow in river nutrition will only increase with time. On the other hand, with continued retreat in glaciation, spring-summer snowfall will melt much faster since it remains longer on the surface of glaciers.

Isotopic composition of the glacier river water in the Mongun-Taiga and Tsambagarav massifs shows less impact of seasonal snow melting on total runoff, because most of the water samples in these massifs are closer to the mean glacier $\delta^{18}$O (Figure 5).

*4.4. Conclusions*

Before isotopic separation, it is necessary to determine what exactly is meant by "glacial runoff". There are two main outlooks on its definition. According to the first, glacial runoff is all water formed as a result of melting on the surface of a glacier [23].

According to the second point of view glacial runoff is the runoff formed due to the melting of long-term reserves of firn and ice. This approach reflects the main hydrological role of glaciers, which consists of the accumulation of precipitation and their temporary exclusion from the water cycle with subsequent redistribution by years.

In this study, glacial runoff is understood only as melting of long-term reserves of firn and ice, which corresponds to the second approach. The choice of this approach is related to the fact that for studied glaciers it is important to understand, how ice and firn reserves influence the formation of runoff in high mountain regions. Seasonal precipitation will feed the mountain rivers even in the absence of glaciers.

Isotopic separation in the Tavan-Bogd massif was made for all streams near the glaciers edge. In the other massifs separation was carried out for main streams, which start in general at the edges of the valley glaciers, because corrie glaciers are not so widespread

and do not produce large streams in these regions. As a clear glacier meltwater isotopic composition, the mean $\delta^{18}O$ of valley glaciers ice was taken. As a seasonal snow meltwater isotopic composition, mean $\delta^{18}O$ of snow from snow pits was used. The results are presented in Table 6.

**Table 6.** Results of glacier runoff isotopic separation for 3 research areas.

| Massif | $\delta^{18}O$ of Ice | $\delta^{18}O$ of Seasonal Snow | Share of the Seasonal Snow, % |
|---|---|---|---|
| Tavan-Bogd | −19.4 | −11.9 | 22–75 |
| Mongun-Taiga | −17.7 | −13.4 | 20 |
| Tsambagarav | −16.7 | −12.6 | 23–24 |

The results of isotopic separation prove suggestions from Section 4.3 of this article: seasonal snow from the surface of glaciers has more of a significant influence on glacier runoff in the Tavan-Bogd massif. The main reasons for this are the better conditions for accumulation of summer snow, according to the widespread occurrence of small corrie glaciers.

It can be concluded that the further degradation of glaciation will have a greater impact on the formation of glacial runoff in Mongun-Taiga and Tsambagarav, than in the Tavan-Bogd massif, where the contribution of seasonal snow is more significant.

In addition, the isotopic characteristics of different objects presented in this manuscript can be used in further hydrological and glaciological research.

**Supplementary Materials:** The Microsoft excel file is available online at https://www.mdpi.com/article/10.3390/w14020252/s1, Table S1: Data which was used for Figure 5 drawing. Raws 4–71—samples from Tavan Bogd, raws 74–155—samples from Mongun-Taiga, raws 158–281—samples from Tsambagarav.

**Author Contributions:** Conceptualization, D.B., D.G. and K.C.; methodology, A.E., I.T. and D.B.; software, A.T.; formal analysis, A.E. and I.T.; investigation, D.B., D.G. and A.T.; writing—original draft preparation, D.B.; writing—review and editing, D.G. and A.T.; visualization, A.T.; supervision, K.C.; project administration, D.G.; funding acquisition, D.G. All authors have read and agreed to the published version of the manuscript.

**Funding:** This research was funded by Russian Foundation for Basic Research (RFBR), grant number No. 19-05-00535.

**Institutional Review Board Statement:** Not applicable.

**Data Availability Statement:** The data of $\delta^2H$ and $\delta^{18}O$ in water and ice presented in this study are available on request from the corresponding author.

**Acknowledgments:** RFBR 19-05-00535 Natural catastrophes and transformation of the landscapes of the southeastern Altai and northwestern Mongolia in the period from the maximum of the last glaciation.

**Conflicts of Interest:** The authors declare no conflict of interest.

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
