# Peer review of "Isotopic Composition of Glacier Ice and Meltwater in the Arid Parts of the Altai Mountains (Central Asia)"

_water, doi:10.3390/w14020252_

Round 1

Reviewer 1 Report

The manuscript water-1522156 presents interesting results on the isotope composition of ice melts in arid central Asia. The work is of sure interest to the journal's readers However, some improvements are necessary. 
In particular:

1) I suggest to include arid central Asia in the title; e.g.: "Isotopic composition of glacier ice and meltwater in the arid 2 parts of the Altai Mountains (arid central Asia)".

2) more up-todate equation of global and local meteoric water lines (and associated bibliographic citations) should be added (see specific comments below)

3) raw data should be included in a supplementary file(s), in particular those used to draw Fig.5 (δ2H are never reported in the tables inserted in the main text).

Specific comments:

line 55-56: I suggest to cite Aizen et al. 2005

line 73: where is that laboratory? St. Petersburg University ? Please specify

line 74: I suggest to write: "by Cavity Ring-Down Spectroscopy (CRDS) using a Picarro L2120-i". Moreover, I suppose that St. Petersburg tap was is a "laboratory standard" (taking into account that the 'official' standard is SMOW). Please specify

line 78-80: using the same spectrometer or other type? Please specify

line 87: (GMWL). For this latter, I suggest to use the more up-to-date equation of Gourcy et al. (2005), which include error on slope and intercept. I copy and paste the equation:

"Long term means (1961-2000) weighted by the amount of precipitation were calculated considering only the years for which more than 70% of precipitation was analysed for a given isotope and at least one year of observation was available. The correlation between the weighted means is:

δ2H = (8.14±0.02)*δ18O + (10.9±0.2), R = 0.98"

Using this latter, the comparison with the best fit of your data (e.g., Fi.g 5) is done in a more effective way (see comment to lines line 201-207).
However, I suggest to use as comparison or the Central Asia Meteoric Water Line (CAMWL) of Yao et al. 2021:

δ2H = 7.30 δ18O + 3.12 (n = 727)

or the equation of Wang et al. (2018, pag. 440):

δD= (7.67 ± 0.08)*δ18O+(3.32±1.15) (n=207)

line 191: where are the deuterium isotope data? I suggest to upload all the raw data used to draw figure 5 in a supplementary file.

line 201-207: in Fig. 5, the blue line is GMWL of a best fit of the samples? Please add the equation of the best fits including error on slope and intercept.

line 290: I suggest to rename/rewrite that section as "Conclusions".

References

Aizen, V. B., Aizen, E., Fujita, K., Nikitin, S. A., Kreutz, K. J., & Takeuchi, L. N. (2005). Stable-isotope time series and precipitation origin from firn-core and snow samples, Altai glaciers, Siberia. Journal of Glaciology, 51(175), 637-654.
https://www.cambridge.org/core/journals/journal-of-glaciology/article/stableisotope-time-series-and-precipitation-origin-from-firncore-and-snow-samples-altai-glaciers-siberia/BA9FF071EF5B4E42BB7AE10C6BC20315

Gourcy, L.L., Groening, M., and Aggarwal, P.K. 2005. Stable oxygen and hydrogen isotopes in precipitation. In: Aggarwal, P.K., Gat, J.R., and Froehlich, K.F.O. (eds.), Isotopes in the Water Cycle: Past, Present and Future of Developing Science. Dordrecht, the Netherlands: Springer, pp. 39–51.
doi: https://doi.org/10.1007/1-4020-3023-1_4

Wang, S., Zhang, M., Hughes, C. E., Crawford, J., Wang, G., Chen, F., ... & Qiu, X. (2018). Meteoric water lines in arid Central Asia using event-based and monthly data. Journal of hydrology, 562, 435-445.
https://doi.org/10.1016/j.jhydrol.2018.05.034

Yao, J., Liu, X., & Hu, W. (2021). Stable isotope compositions of precipitation over Central Asia. PeerJ, 9, e11312.
https://www.ncbi.nlm.nih.gov/pmc/articles/PMC8088211/

Hope this helps

Author Response

Dear Reviewer! I've added my answer in the word doc below. 

Thank you!

Reviewer 2 Report

The revised article is on the studies of  the isotopic composition of glacier ices and meltwaters  for region of Altai Mountains. Study area comprised of three 3 massifs within distance of about 200 km from ich other.  Authors analysed two stable isotope indicators of d2H and d18O with more attention being paid to discussing the results related to isotopes of oxygen. All samples were collected in  the summer time.  

I feel the manuscript is publishable. Following comments should be considered in the revision:

  1. Did authors try to analyse the d18O of snow precipitation in the winter time. If not why?
  2. What is the d18O of precipitations?
  3. Did author make studies of older glacier ices (by studies of ice cores taken from boreholes)?

Minor notes:

Line 65 delta value not letter

Line 87,206 check abbreviation

Line 118 rewrite “is varies”

Line 189 English, please.

Figure 5 Please add the plot of GMWL to each Figure -  for comparison purpose.

Line 214 Rewrite sentence, please.

Line 236 Maybe not all readers will know what is the reason for such situation, please add the explanation why on west part the precipitation is higher.

Author Response

Dear Reviewer! I've added my answer in the word file below.

Thank you!

Round 2

Reviewer 1 Report

The authors have satisfactorily responded to all my questions and comments

Author Response

thank you

Reviewer 2 Report

The authors have corrected the manuscript according to my suggestions and it may now be published.

Author Response

thank you